# Variable Gain Prescribed Performance Control for Dynamic Positioning of Ships with Positioning Error Constraints

Chenglong Gong 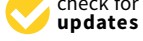, Yixin Su * and Danhong Zhang

School of Automation, Wuhan University of Technology, Wuhan 430070, China; gongcl@whut.edu.cn (C.G.); zhangdh@whut.edu.cn (D.Z.)
* Correspondence: suyixin@whut.edu.cn; Tel.: +86-13627298717

**Abstract:** In this paper, a variable gain prescribed performance control law is proposed for dynamic positioning (DP) of ships with positioning error constraints, input saturation and unknown external disturbances. The error performance index functions are designed to preset the prescribed performance bounds and the error mapping functions are constructed to incorporate the prescribed performance bounds into the DP control design. The variable gain technique is used to limit the output amplitude of the control law to avoid input saturation of the system by dynamically adjusting the control gain of the DP control law according to the positioning errors, and the error mapping function replaces the positioning error as a recursive sliding-mode surface to realize the prescribed performance control of the system and guarantee the stability of the closed-loop system with variable control gains. It has been proved that the proposed DP control law can make the uniformly ultimately boundedness of all signals in the DP closed-loop control system. The numerical simulation results illustrate that the proposed control law can make the ship's position and heading maintain at the desired value with positioning error constraints, enhance the non-fragility of the DP control law to the perturbation of system's parameters and improve the system's rejection ability to external disturbances.

**Keywords:** dynamic positioning of ships; variable gain technique; recursive sliding-mode; prescribed performance; input saturation; positioning error constraints



## 1. Introduction

The dynamic positioning (DP) technique is widely used in ship's positioning and motion control [1]. The ship's DP means that the ship controls its own propulsion system to resist the external disturbances, so that the ship can reach and maintain the desired position or track the reference trajectory with a certain attitude [2,3]. Compared with the traditional anchor moored positioning, the DP mode has the advantages of strong flexibility, high positioning accuracy and low positioning costs. In practical engineering, the DP ship will execute some accurate control tasks [4–8], such as the fixed-point salvage, parallel navigation of multiple ships, collision avoidance of offshore platforms, underwater engineering construction and underwater vehicle tracking. Considering the actual size of the DP ship, the ship is required to strictly follow the planed path or accurately locate at the desired position, so as to avoid collisions caused by excessive position errors when the ship passes through a specific area or unknown area. Therefore, accurate DP control plays an important role in the ship's practical engineering.

With the development of the nonlinear control theory, the DP control technique has been greatly developed and a series of DP methods have been proposed [9–13]. However, the DP system is a complex nonlinear system with multi-inputs and multi-outputs (MIMO), which is easily affected by external disturbances. Vaerno et al. [14] designed a model-based disturbance observer to estimate the external disturbances acting on the DP ship. Brodtkorb et al. [15] developed an online estimation method for unknown disturbances to design a DP controller, which realized the feedforward compensation of wave

disturbances. Yang et al. [16] estimated and compensated the unknown time-varying disturbances by designing a disturbance observer and realized the accurate trajectory tracking of the DP ship. Hu et al. [17] constructed an adaptive disturbance observer to obtain the unknown disturbance estimation, and it doesn't need any prior information of the ship's mathematical model. Considering unknown disturbances and the perturbation of the ship's mathematical model parameters, Qu et al. [18] designed an exponentially stable backstepping controller to estimate the composite disturbances. Zhao et al. [19] used an adaptive compensator to reduce the effects of external disturbances on the system, and developed a terminal sliding-mode DP control law for offshore platforms.

Input saturation is an inevitable nonlinear factor affecting the control performance of the system. Some effective methods for dealing with input saturation have been proposed. Guerreiro et al. [20] transformed input saturation constraints into optimization conditions and designed a model predictive controller to eliminate the adverse effects of input saturation on the system. Sarhadi et al. [21] designed an adaptive PID control law based on the anti-saturation compensator for underwater vehicles with unknown mathematical model parameters and input saturation. Chang et al. [22] developed a parallel distributed compensation fuzzy DP controller to handle input saturation and external disturbances. Hu et al. [23] designed a robust adaptive DP control law to deal with input saturation by constructing an auxiliary dynamic system.

Most DP control algorithms are designed with the dynamic surface control (DSC) method. However, the design of the DSC method is based on the linear gain [24], which makes the contradiction between the dynamic quality and the control accuracy of the system. When the control gain is too large, the control accuracy of the system is high, but the system's input saturation will occur, and when the control gain is too small, the control accuracy of the system will decrease. Moreover, the design of the DSC method depends on local errors of the system, which makes the control system robust to system's uncertainties, but the control performance of the controller is easily affected by the perturbation of the system's parameters. Sliding-mode control [25–31] has strong robustness to unmodeled dynamics of nonlinear systems. Liu et al. [32] constructed a nonlinear gain function and proposed an improved DSC strategy with sliding-mode control for a class of nonlinear systems to enhance the non-fragility of the control law. Shen et al. [33] considered the relationship between the ship's position errors and velocity errors to design a recursive sliding-mode trajectory tracking control law to ensure the boundedness of all signals in the closed-loop control system. Shao et al. [34] proposed an adaptive recursive terminal sliding-mode controller to improve the control performance of the linear motor.

The prescribed performance control has been successfully applied to nonlinear systems. Wang et al. [35] used a non-logarithmic piecewise error mapping function to design a prescribed performance control law for a class of MIMO nonlinear systems. Dai et al. [36] developed an adaptive neural network control law based on the error transform function for ships with the inaccuracy mathematical model to guarantee the preset tracking performance of the system. Li et al. [37] proposed a robust adaptive prescribed performance control for DP ships to make the ship's position and heading maintain at the desired value with the prescribed performance requirements. Wang et al. [38] designed a fuzzy DP controller based on the integrating prescribed performance control and backstepping control for ships to make the trajectory tracking errors strictly comply within prescribed performance envelopes.

Considering the positioning error constraints, input saturation and unknown external disturbances, a variable gain prescribed performance control law for DP ships is proposed to ensure that all signals in the DP closed-loop control system are uniformly ultimately bounded and guarantee that the positioning errors meet the prescribed performance requirements without input saturation of the system. The main contributions of this paper are as follows.

(1)   An improved DP control law is proposed for ships to prevent input saturation of the system and deal with positioning error constrains in the same control framework.

The variable gain technique is used to dynamically adjust the control gain of the DP control law according to positioning errors to alleviate the contradiction between the dynamic quality and the control accuracy of the DP system with input saturation constrains. The error performance index functions and error mapping functions are designed to ensure that the ship's DP meets the prescribed performance requirements.

(2) The error mapping function replaces the positioning error as a recursive sliding-mode surface to realize the prescribed performance control of the system and guarantee the stability of the closed-loop system with variable control gains, then an improved recursive sliding-mode control is designed based on the DSC to enhance the non-fragility of the DP control law to the perturbation of system's parameters.

The remaining structure of this paper is arranged as follows. The problem formulation is presented in Section 2, the DP control design is shown in Section 3, the stability analysis is detailed in Section 4, the simulations and the conclusion are presented in Sections 5 and 6, respectively.

**Notation:** In this paper, $\Re^{m \times n}$ defines all $m \times n$ real matrices; $\lambda_{\min}(\cdot)$ represents the minimum eigenvalue of the matrix; $\|\cdot\|$ is the Euclidean norm of the matrix; $diag(\cdot)$ stands for the diagonal matrix; $R$ represents the set of real numbers.

## 2. Problem Formulation

Assume that the ship is bilaterally symmetric. The ship's motion coordinate frame is shown in Figure 1, where $OX_0Y_0$ is an inertial coordinate frame and the axes $OX_0$ and $OY_0$ are directed to north and east, respectively. $AXY$ is a body-fixed coordinate frame, where $A$ is the ship's center of gravity, and axes $AX$ and $AY$ are directed to fore and starboard, respectively.

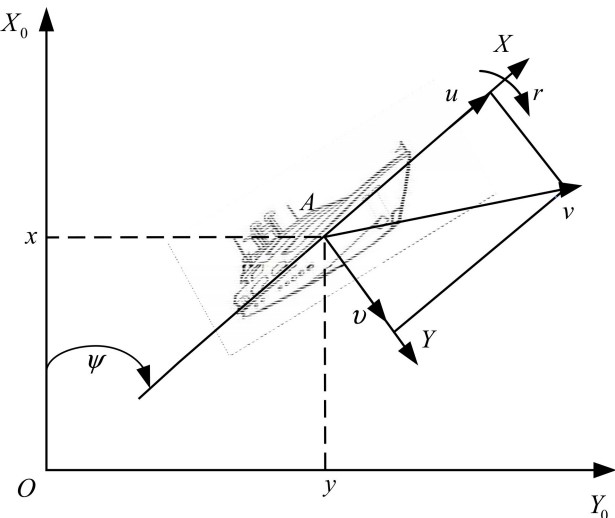

**Figure 1.** The ship's motion coordinate frame.

The motion mathematical model of the DP ship is expressed as

$$\dot{\eta} = J(\psi)v, \tag{1}$$

$$M\dot{v} + Dv = \tau + d(t), \tag{2}$$

where $\eta = [x, y, \psi]^T$ is the ship's position vector composed of the position $(x, y)$ and heading $\psi \in [0, 2\pi]$; $v = [u, v, r]^T$ is the ship's velocity vector in the body-fixed coordinate system, which is composed of the surge velocity $u$, the sway velocity $v$ and the yaw rate $r$; $J(\psi) \in \Re^{3 \times 3}$ is the rotation matrix with the properties of $J^T(\psi) = J^{-1}(\psi)$ and $\|J(\psi)\| = 1$, it can be expressed as

$$J(\psi) = \begin{bmatrix} \cos(\psi) & -\sin(\psi) & 0 \\ \sin(\psi) & \cos(\psi) & 0 \\ 0 & 0 & 1 \end{bmatrix}. \tag{3}$$

$M \in \Re^{3\times3}$ is the inertia matrix; $D \in \Re^{3\times3}$ is the linear damping matrix; $\tau = [\tau_x, \tau_y, \tau_\psi]^T$ represents the equivalent control vector composed of the equivalent forces and moment provided by the ship's propulsion system, and $\tau_x, \tau_y, \tau_\psi$ are the surge force, sway force and yaw moment, respectively; $d(t) = [d_1(t), d_2(t), d_3(t)]^T$ is the external disturbance vector caused by wind, waves, and currents.

Due to the physical limitation of the propeller, the ship's equivalent control forces and moment provided by the propulsion system are limited, i.e.,

$$|\tau_k| \le \tau_{M,k}, k = x, y, \psi, \tag{4}$$

where $\tau_{M,k}, k = x, y, \psi$ are input saturation amplitudes.

To facilitate the design of the DP control law and the analysis of the system's stability, the following assumption and lemma are made.

**Assumption 1.** *The external disturbances $d_i(t), i = 1, 2, 3$ are unknown, time-varying yet bounded, and satisfy*

$$\left\| \dot{d}(t) \right\| \le \rho < \infty, \tag{5}$$

*where $\rho$ is a positive constant.*

**Lemma 1.** *Assume that the positive definite function $V(t)$ is continuously differentiable on $[t_0, +\infty)$, $w(t)$ and $l(t)$ are continuous. If $V(t_0) \le G$ for $t \ge t_0$, where $G \in R$, and satisfy*

$$\dot{V}(t) \le V(t)l(t) + w(t), \tag{6}$$

*then, for $t \ge t_0$, there is*

$$V(t) \le (G + \int_{t_0}^t w(s)e^{\int_{t_0}^s -l(\tau)\mathrm{d}\tau}\mathrm{d}s)e^{\int_{t_0}^s l(\tau)\mathrm{d}\tau}. \tag{7}$$

Specially, when $w(t) = A$, $l(t) = K$ and $t_0 = 0$, for $t \ge 0$, there is

$$V(t) \le -\frac{A}{K} + (G + \frac{A}{K})e^{Kt}. \tag{8}$$

Since the ocean environment is time-varying, unpredictable and has finite energy, the external disturbances acting on the ship can be regarded as the unknown, time-varying and bounded signals with finite changing rates, so Assumption 1 is reasonable.

In the process of the ship's DP, the proposed DP control law provides the equivalent forces and moment needed by the ship to suppress the external disturbances, and then the thrust distribution unit calculates them as the command signals of each propeller in the propulsion system, such as the propeller rotation speed, the direction angle, the rudder angle, etc. Finally, the control forces and moment required by the control law are generated by the coordinating action of each propeller, so that the ship can position at the desired value or track the reference trajectory with a certain attitude.

The objective of this paper is to propose a DP control law $\tau$ for ships with positioning error constraints, input saturation and unknown external disturbances to make the ship's position and heading maintain at the desired value with positioning error constraints, all signals in the DP closed-loop control system are uniformly ultimately bounded and the output amplitudes of the control law meet $|\tau_k| \le \tau_{M,k}, k = x, y, \psi$.

## 3. DP Control Design

For the ship's motion mathematical model (1) and (2), we introduce a nonlinear disturbance observer [23] to estimate and compensate the external disturbances in this section, and design the error performance index functions and error mapping functions to make the ship's DP meet the prescribed performance requirements, then design a variable gain recursive sliding-mode DSC strategy to prevent input saturation and improve the dynamic quality of the system. Finally, a variable gain DP control law with the prescribed performance is proposed.

### 3.1. The Nonlinear Disturbance Observer

For the unknown time-varying disturbances $d$, a nonlinear disturbance observer is introduced, which can be described as

$$\begin{cases} \hat{d} = \beta + K_0 M v \\ \dot{\beta} = -K_0 \beta - K_0[-Dv + \tau + K_0 M v] \end{cases},\tag{9}$$

where $\hat{d}$ is the estimated value of the disturbances; $K_0 \in \Re^{3 \times 3}$ is the positive definite symmetric observer gain matrix; $\beta \in \Re^{3 \times 1}$ is the auxiliary vector of the observer.

Define the disturbance estimation error vector as $\tilde{d} = \hat{d} - d$, and according to (2) and (9), we obtain

$$\begin{aligned} \dot{\hat{d}} &= -K_0[\beta + K_0 M v - d] \\ &= -K_0 \tilde{d} \end{aligned}\tag{10}$$

The disturbance observer (9) can provide the accurate estimation vector $\hat{d}$ for the external disturbance vector $d$, rather than the estimation of constant upper bounds of the disturbances. As a result, the ship can reduce the energy consumption in the DP process. In addition, the disturbance observer is exponentially stable, and its stability will be proved in the latter discussion.

### 3.2. A Variable Gain Function

To limit the output amplitude of the DP control law within the input saturation amplitude range of the propulsion system, a variable gain function is designed as

$$\lambda(x) = a|x|^{\frac{1}{b}} \text{sgn}(x),\tag{11}$$

where $a > 0$; $b$ is a positive integer. The variable gain function has the characteristic of "Large gains for small errors, small gains for large errors", and has the following properties.

**Property 1.** *The function $\lambda(x)$ strictly monotonically increases with respect to the independent variable x. In order to avoid the excessive slope of $\lambda(x)$ when $x = 0$, define*

$$\frac{d\lambda(x)}{dx} = \begin{cases} 1, & x = 0 \\ \frac{a}{b}|x|^{\frac{1-b}{b}}, & x \neq 0 \end{cases}\tag{12}$$

**Property 2.** *Define*

$$\hbar(x) = \frac{1}{2}\left[\frac{d\lambda(x)}{dx}x + \lambda(x)\right],\tag{13}$$

*for$\forall x \in R$, then*

$$x \times \hbar(x) = \frac{1}{2}\left[\frac{d\lambda(x)}{dx}x^2 + x \times \lambda(x)\right] \geq \frac{1}{2}x \times \lambda(x).\tag{14}$$

**Property 3.** *Define*

$$\gamma(x) = \begin{cases} 1, & x = 0 \\ \frac{\hbar(x)}{x}, & x \neq 0 \end{cases}, \tag{15}$$

*for $\forall x \in R$, then*

$$\gamma(x) = \frac{1}{2}\left[\frac{d\lambda(x)}{dx} + \frac{\lambda(x)}{x}\right] > 0. \tag{16}$$

Taking different values of $b$ when $a = 0.5$, the change curves of $\lambda(x)$ are shown in Figure 2. It can be seen that the slope of curves decreases with the increase of $|x|$.

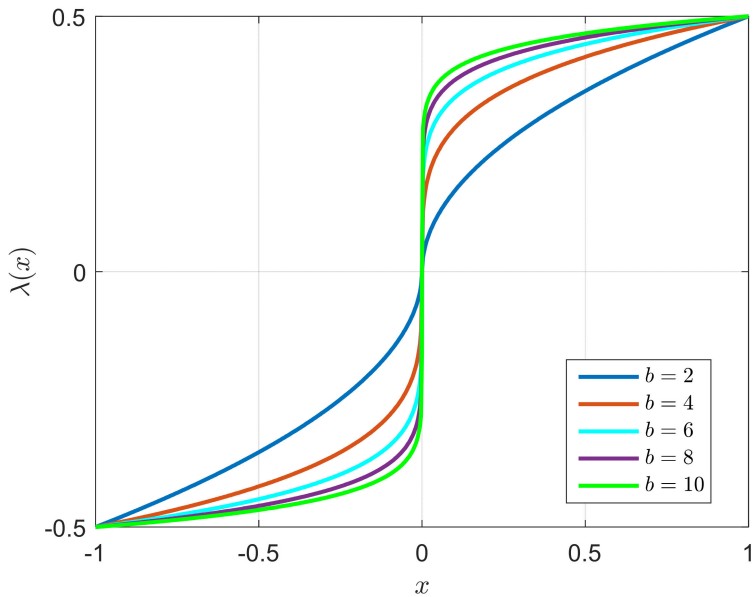

**Figure 2.** The change curves of $\lambda(x)$.

### 3.3. The Prescribed Performance Control Design

The prescribed performance control technique can make the tracking error of the closed-loop system converge to a preset allowable range, while ensuring that the convergence rate meets the preseted requirements, thereby improving the control performance of the system.

In the DP task, the DP ship should reach and maintain at the desired value $\eta_d = [\eta_{d,x}, \eta_{d,y}, \eta_{d,\psi}]^T$ with positioning error constraints, which means that the positioning error should satisfy

$$-\delta_k \rho_{m,k}(t) < e_k(t) < \delta_k \rho_{M,k}(t), \forall t \geq 0, k = x, y, \psi, \tag{17}$$

with

$$\rho_{M,k}(t) = (\rho_{M,k}(0) - \rho_{M,k}(\infty))e^{-\mu_k t} + \rho_{M,k}(\infty), \tag{18}$$

$$\rho_{m,k}(t) = (\rho_{m,k}(0) - \rho_{m,k}(\infty))e^{-\mu_k t} + \rho_{m,k}(\infty), \tag{19}$$

where $e = \eta - \eta_d = [e_x, e_y, e_\psi]^T$ is the positioning error vector; $\rho_{M,k}(t)$ and $\rho_{m,k}(t)$ are error performance index functions and they are strictly decreasing; $\rho_{M,k}(0), \rho_{M,k}(\infty), \rho_{m,k}(0), \rho_{m,k}(\infty)$, $\mu_k$ and $\delta_k$ are positive constants and $\rho_{m,k}(\infty) < \rho_{m,k}(0), \rho_{M,k}(\infty) < \rho_{M,k}(0), 0 < \delta_k \leq 1$.

Theoretically, it is difficult to directly incorporate the prescribed performance bounds defined by (17)–(19) into the DP control design, so the error mapping functions are constructed to resolve the above problem, which can be expressed as

$$E_{1,k} = \frac{e_k}{(\rho_{M,k} - e_k)(e_k + \rho_{m,k})}, k = x, y, \psi. \tag{20}$$

Moreover, $E_{1,k}, k = x, y, \psi$ are strictly increasing, and under the following condition

$$\rho_{m,k}(0) < e_k(0) < \rho_{M,k}(0), \tag{21}$$

the boundedness of $E_{1,k}$ for all $t \geq 0$ is sufficient to ensure that the positioning error meets the requirements of (17) when $0 < \delta_k \leq 1$.

*3.4. Variable Gain Recursive Sliding-Mode Prescribed Performance DP Control Design*

Based on the variable gain technique, the prescribed performance control and DSC, an improved recursive sliding-mode control is designed to enhance the non-fragility of the control law to the perturbation of system's parameters and resolve the stability proving problem of the system with variable control gains. Then, combining the nonlinear disturbance observer, the variable gain prescribed performance DP control law is proposed.

To realize the prescribed performance control of the system and guarantee the stability of the closed-loop system with variable control gains, the error mapping function vector $E_1$ replaces the positioning error vector $e$ as the first recursive sliding-mode surface vector $s_1 \in \Re^{3 \times 1}$, and it can be described as

$$s_1 = E_1, \tag{22}$$

where $E_1 = [E_{1,x}, E_{1,y}, E_{1,\psi}]^T$.

Taking the derivative of (22), we obtain

$$\dot{s}_1 = \Pi J(\psi) v + Q, \tag{23}$$

where $\Pi = diag(\Pi_x, \Pi_y, \Pi_\psi), Q = [Q_x, Q_y, Q_\psi]^T, Q_k = \frac{-e_k^2(\dot{\rho}_{M,k} - \dot{\rho}_{m,k}) + e_k(\dot{\rho}_{M,k}\rho_{m,k} + \rho_{M,k}\dot{\rho}_{m,k})}{(\rho_{M,k} - e_k)^2(e_k + \rho_{m,k})^2}$, $\Pi_k = \frac{e_k^2 + \rho_{M,k}\rho_{m,k}}{(\rho_{M,k} - e_k)^2(e_k + \rho_{m,k})^2} > 0, k = x, y, \psi$.

Design the virtual control vector $\alpha_1 \in \Re^{3 \times 1}$ as

$$\alpha_1 = J^T(\psi)\left(\Pi^{-1}(-K_1 \hbar_1(s_1) - Q - C_1 s_1) - \Pi \hbar_1(s_1)\right) + C_2 s_1, \tag{24}$$

where $K_1, C_1, C_2$ are $3 \times 3$ symmetric positive definite matrices; $\hbar_1(s_1) = [\hbar(s_{1,1}), \hbar(s_{1,2}), \hbar(s_{1,3})]^T \in \Re^{3 \times 1}$ is a variable gain function vector.

The first order low-pass filter is designed as

$$T\dot{v}_d + v_d = \alpha_1, v_d(0) = \alpha_1(0), \tag{25}$$

where $T > 0$ is the filter time constant; $v_d \in \Re^{3 \times 1}$ is the low-pass filter state vector.

Inspired by the idea of DSC strategy, the state vector $v_d$ is introduced into the design of the recursive sliding-mode control. It can be seen from (25) that the differential term $\dot{v}_d$ is directly obtained from $\dot{v}_d = (\alpha_1 - v_d)/T$, which can effectively avoid the complicated derivative calculation of $\alpha_1$ in the backstepping control and greatly simplify the design process of the DP control law.

Define the filtering error vector of the system as

$$Y = v_d - \alpha_1. \tag{26}$$

The velocity error vector is defined as

$$z_2 = v - v_d. \tag{27}$$

The second recursive sliding-mode surface vector $s_2 \in \Re^{3 \times 1}$ is defined as

$$s_2 = C_2 s_1 + z_2. \tag{28}$$

In view of (2), (27) and (28), we obtain

$$M\dot{s}_2 = MC_2\dot{s}_1 + d - Dv + \tau - M\dot{v}_d. \tag{29}$$

Design the variable gain prescribed performance DP control law as

$$\tau = Dv + M\dot{v}_d - MC_2\dot{s}_1 - K_2\hbar_2(s_2) - C_3s_2 - N_2(s_2)J^T(\psi)\Pi\hbar_1(s_1) - \hat{d}, \tag{30}$$

where $\hbar_2(s_2) = [\hbar(s_{2,1}), \hbar(s_{2,2}), \hbar(s_{2,3})]^T \in R^3$; $K_2, C_3 \in \Re^{3\times3}$ are positive definite symmetric matrices; $N_2(s_2) = diag(1/\gamma_2(s_{2,1}), 1/\gamma_2(s_{2,2}), 1/\gamma_2(s_{2,3}))$.

In the DP control law (30), the variable gain function vector $\hbar_2(s_2)$ with variable gain properties can effectively alleviate the contradiction between the control accuracy and dynamic quality of the DP system; the term $N_2(s_2)J^T(\psi)\Pi\hbar_1(s_1)$ is used to eliminate the coupling of the system.

The block diagram of the DP closed-loop control system is shown in Figure 3.

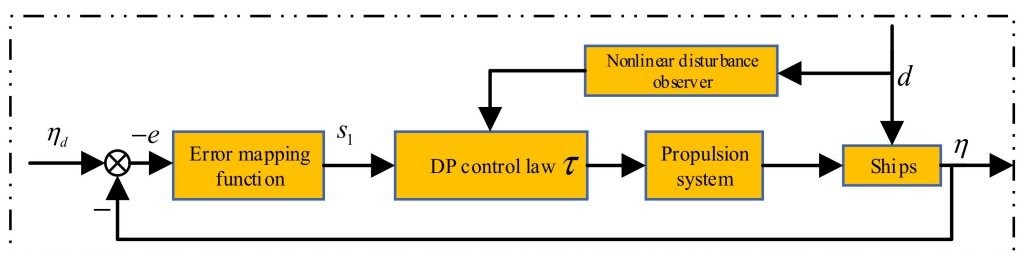

**Figure 3.** The block diagram of the DP closed-loop control system.

## 4. Stability Analysis

Define the Lyapunov function as

$$V = V_d + V_c. \tag{31}$$

Select $V_d$ as

$$V_d = \frac{1}{2}\tilde{d}^T\tilde{d}. \tag{32}$$

In view of (9), (10) and *Young's* inequality, the derivative of (32) can be obtained as

$$\begin{aligned}\dot{V}_d &= \tilde{d}^T\dot{\tilde{d}} \\ &= \tilde{d}^T(-K_0\tilde{d} - \dot{d}) \\ &\leq -\tilde{d}^TK_0\tilde{d} + \tilde{d}^T\tilde{d} + \frac{1}{4}\rho^2 \\ &\leq -\mu_dV_d + C_d\end{aligned} \tag{33}$$

where $\mu_d = 2(\lambda_{\min}(K_0) - 1)$ and $K_0$ satisfies $\lambda_{\min}(K_0) > 1$; $C_d = \frac{1}{4}\rho^2$.

**Theorem 1.** *The estimation error$\tilde{d}$ of the nonlinear disturbance observer (9) can reach and remain in the arbitrarily small bounded compact set$\Omega_{\tilde{d}} = \left\{\tilde{d} \in R^3\middle|\left\|\tilde{d}\right\| \leq \zeta_{\tilde{d}}, \zeta_{\tilde{d}} > \sqrt{C_d/\mu_d}\right\}$ by properly selecting the design matrix$K_0$.*

Solving (33), we obtain

$$0 \leq V_d(t) \leq \frac{C_d}{\mu_d} + \left[V_d(0) - \frac{C_d}{\mu_d}\right]e^{-\mu_dt}. \tag{34}$$

Furthermore, according to (32) and (34), we have

$$0 \leq \left\| \tilde{d} \right\| \leq \sqrt{\frac{2C_d}{\mu_d} + 2\left[V_d(0) - \frac{C_d}{\mu_d}\right] e^{-\mu_d t}}. \tag{35}$$

According to Lemma 1 and (34), we know that $V_d(t)$ is bounded, and it can be seen from (35) that for any $\zeta_{\tilde{d}} > \sqrt{C_d/\mu_d}$, there exists a time constant $T_{\tilde{d}} > 0$ such that $\left\| \tilde{d} \right\| \leq \zeta_{\tilde{d}}$ for all $t > T_{\tilde{d}}$, i.e., $\left\| \tilde{d} \right\|$ settles within $\Omega_{\tilde{d}} = \left\{ \tilde{d} \in R^3 \left| \left\| \tilde{d} \right\| \leq \zeta_{\tilde{d}}, \zeta_{\tilde{d}} > \sqrt{C_d/\mu_d} \right. \right\}$, it means that the compact set $\Omega_{\tilde{d}}$ can be made arbitrarily small by properly selecting the design matrix $K_0$. Hence, Theorem 1 is proved.

Choose $V_c$ as

$$V_c = \frac{1}{2}\lambda_1{}^T(s_1)s_1 + \frac{1}{2}\lambda_2{}^T(s_2)Ms_2 + \frac{1}{2}Y^T Y. \tag{36}$$

Taking the derivative of (36), we have

$$\dot{V}_c = \hbar_1{}^T(s_1)\dot{s}_1 + \hbar_2{}^T(s_2)M\dot{s}_2 + Y^T\dot{Y}. \tag{37}$$

In view of (26), (27) and (28), we obtain

$$\begin{aligned}
\hbar_1{}^T(s_1)\dot{s}_1 &= \hbar_1{}^T(s_1)(Q + \Pi J(\psi)(s_2 + Y + \alpha_1 - C_2 s_1)) \\
&= \hbar_1{}^T(s_1)\big(\Pi J(\psi)(s_2 + Y) - K_1\hbar_1(s_1) - C_1 s_1 - \Pi^2\hbar_1(s_1)\big) \cdot
\end{aligned} \tag{38}$$

According to *Young's* inequality, we have

$$\hbar_1{}^T(s_1)\Pi J(\psi)Y \leq \hbar_1{}^T(s_1)\Pi^2\hbar_1(s_1) + \frac{1}{4}Y^T Y. \tag{39}$$

In the light of (38) and (39), we have

$$\hbar_1{}^T(s_1)\dot{s}_1 \leq \hbar_1{}^T(s_1)\Pi J(\psi)s_2 + \frac{1}{4}Y^T Y - \hbar_1{}^T(s_1)K_1\hbar_1(s_1) - \hbar_1{}^T(s_1)C_1 s_1. \tag{40}$$

For the term $\hbar_2{}^T(s_2)M\dot{s}_2$, according to (29), (30), *Young's* inequality and $\tilde{d} = \hat{d} - d$, we obtain

$$\begin{aligned}
\hbar_2{}^T(s_2)M\dot{s}_2 &= \hbar_2{}^T(s_2)[-K_2\hbar_2(s_2) - C_3 s_2 - N_2(s_2)J^T(\psi)\Pi\hbar_1(s_1) - \tilde{d}] \\
&\leq -\hbar_2{}^T(s_2)K_2\hbar_2(s_2) - \hbar_2{}^T(s_2)C_3 s_2 - \hbar_2{}^T(s_2)N_2(s_2)J^T(\psi)\Pi\hbar_1(s_1) \\
&\quad + \hbar_2{}^T(s_2)\hbar_2(s_2) + \frac{1}{4}\tilde{d}^T\tilde{d}
\end{aligned} \tag{41}$$

For the term $Y^T\dot{Y}$, according to (24) and (26), we know that $Y$ is a function of $(s_1, \Pi, Q)$, so

$$\begin{aligned}
\dot{Y} &= \dot{v}_d - \dot{\alpha}_1 \\
&= -\frac{Y}{T} - \frac{\partial\alpha_1}{\partial\Pi}\dot{\Pi} - \frac{\partial\alpha_1}{\partial Q}\dot{Q} - \frac{\partial\alpha_1}{\partial s_1}\dot{s}_1
\end{aligned} \tag{42}$$

Considering the compact sets $\Omega_e = \left\{ \left[\Pi^T, \dot{\Pi}^T, Q^T, \dot{Q}^T\right]^T : \|\Pi\|^2 + \left\|\dot{\Pi}\right\|^2 + \|Q\|^2 + \left\|\dot{Q}\right\|^2 \leq \beta_0 \right\}$, and $\Omega_1 = \left\{ \left[s_1^T, s_2^T, Y^T\right]^T : V \leq \varpi_0 \right\}$, where $\varpi_0$ and $\beta_0$ are positive constants, which means that $\Omega_1 \times \Omega_e$ is also a compact set, so there is a non-negative con-

tinuous function $B(\cdot)$, such that $\left\|\dot{Y} + Y/T\right\| \leq B(s_1, s_2, Y, \Pi, \dot{\Pi}, Q, \dot{Q})$. And there exists a maximum value $B_M$ for $\|B(\cdot)\|$ over $\Omega_1 \times \Omega_e$, i.e., $\|B(\cdot)\| \leq B_M$, then we have

$$
\begin{aligned}
Y^T\dot{Y} &= -\frac{Y^TY}{T} + Y^T\left(\frac{Y}{T} + \dot{Y}\right) \\
&\leq -\frac{Y^TY}{T} + \|Y\|^2\|B(\cdot)\|^2 + \frac{1}{4} \\
&\leq -\frac{Y^TY}{T} + B_M^2\|Y\|^2 + \frac{1}{4}
\end{aligned}
\tag{43}
$$

Considering

$$
\hbar_1^T(s_1)\Pi J(\psi)s_2 = \hbar_2^T(s_2)N_2(s_2)J^T(\psi)\Pi\hbar_1(s_1),
\tag{44}
$$

$$
\hbar_1^T(s_1)C_3 s_1 \geq \frac{1}{2}\lambda_1^T(s_1)C_3 s_1,
\tag{45}
$$

$$
\hbar_2^T(s_2)C_2 s_2 \geq \frac{1}{2}\lambda_2^T(s_2)C_2 s_2,
\tag{46}
$$

and according to (14), (40), (41), (44)–(46) and *Young's* inequality, we have

$$
\begin{aligned}
\dot{V}_c \leq\ & -\lambda_{\min}(K_1)\hbar_1^T(s_1)\hbar_1(s_1) - \hbar_1^T(s_1)C_1 s_1 - [\lambda_{\min}(K_2) - 1]\hbar_2^T(s_2)\hbar_2(s_2) \\
& -\hbar_2^T(s_2)C_3 s_2 - \left[\frac{1}{T} - \frac{1}{4}\right]Y^TY + \frac{1}{4}\tilde{d}^T\tilde{d} + B_M^2\|Y\|^2 + \frac{1}{4}
\end{aligned}
\tag{47}
$$

When $\lambda_{\min}(K_2) > 1$, in view of (45)–(47), we have

$$
\begin{aligned}
\dot{V}_c &\leq -\lambda_{\min}(C_1)\lambda_1^T(s_1)s_1 - \frac{\lambda_{\min}(C_3)}{\lambda_{\min}(M)}\lambda_2^T(s_2)Ms_2 - \left(\frac{1}{T} - \frac{1}{4}\right)Y^TY + \frac{1}{4}\tilde{d}^T\tilde{d} + B_M^2\|Y\|^2 + \frac{1}{4} \\
&\leq -\mu_c V_d + C_V
\end{aligned}
\tag{48}
$$

where $\mu_c = \min\left\{2\lambda_{\min}(C_1), 2\frac{\lambda_{\min}(C_3)}{\lambda_{\max}(M)}, 2\left(\frac{1}{T} - \frac{1}{4}\right), \frac{1}{2}\right\} > 0$, $C_V = B_M^2\|Y\|^2 + \frac{1}{4}$. Combining (33) and (48), we obtain

$$
\begin{aligned}
\dot{V} &\leq -\lambda_{\min}(C_1)\lambda_1^T(s_1)s_1 - \frac{\lambda_{\min}(C_3)}{\lambda_{\min}(M)}\lambda_2^T(s_2)Ms_2 - \left(\frac{1}{T} - \frac{1}{4}\right)Y^TY \\
&\quad -\left[\lambda_{\min}(K_0) - \frac{5}{4}\right]\tilde{d}^T\tilde{d} + B_M^2\|Y\|^2 + \frac{1}{4} + \frac{1}{4}\rho^2 \\
&\leq -\mu V + C
\end{aligned}
\tag{49}
$$

where $\mu = 2\min\left\{\lambda_{\min}(C_1), \frac{\lambda_{\min}(C_3)}{\lambda_{\max}(M)}, (\lambda_{\min}(K_0) - \frac{5}{4}), \left(\frac{1}{T} - \frac{1}{4}\right)\right\} > 0$, $C = B_M^2\|Y\|^2 + \frac{1}{4} + \frac{1}{4}\rho^2$.

To make $V \leq \varpi_0$, the following conditions must be satisfied

$$
\frac{1}{T} - \frac{1}{4} > 0,
\tag{50}
$$

$$
\lambda_{\min}(K_0) > \frac{5}{4}, \ \lambda_{\min}(K_2) > 1.
\tag{51}
$$

Solving (49), we have

$$
0 \leq V(t) \leq \frac{C}{\mu} + \left[V(0) - \frac{C}{\mu}\right]e^{-\mu t}.
\tag{52}
$$

According to Lemma 1 and (52), we know that $V(t)$ is uniformly ultimately bounded. Therefore, according to (31), (32) and (36), $\|s_1\|, \|s_2\|, \|Y\|, \left\|\tilde{d}\right\|$ are bounded. In view of (17)–(19) and $e = \eta - \eta_d$, $\|\eta\|$ is bounded. Additionally, in the light of (24)–(27), $\|\alpha_1\|, \|v\|, \|v_d\|$ are bounded. Hence, all signals in the DP closed-loop control system are uniformly ultimately bounded.

According to (36) and (52), we have

$$0 \leq \|s_1\| \leq \sqrt{\frac{2C}{\mu} + 2\left[V(0) - \frac{C}{\mu}\right]e^{-\mu t}}. \tag{53}$$

It can be seen from (53) that for any $\zeta \geq \sqrt{2C/\mu}$, there exists a positive constant $t_\zeta > 0$ such that $\|s_1\|$ settles within $\Omega_{s_1} = \left\{s_1 \in R^3 \middle| \|s_1\| \leq \zeta\right\}$ for all $t > t_\zeta$ so that the compact set $\Omega_{s_1}$ can be made arbitrarily small by properly selecting $\delta_k, \rho_{M,k}(0), \rho_{M,k}(\infty), \rho_{m,k}(0), \rho_{m,k}(\infty)$, $k = x, y, \psi$ and $T, a, b, K_0, K_1, K_2, C_i, i = 1, 2, 3$. Therefore, according to (14)–(17) and $e = \eta - \eta_d$, the ship's position $(x, y)$ and heading $\psi$ can reach and maintain at the $\eta_d = [x_d, y_d, \psi_d]^T$ with the positioning error meeting the prescribed performance requirements.

## 5. Simulations

In this section, we will use the variable gain prescribed performance DP control law $\tau$ to simulate a supply ship DP task, and compare it with the state feedback DP control law $\tau_f$ and robust nonlinear DP control law $\tau_c$ in two different cases. The motion mathematical model parameters of the supply ship are detailed in [39]. The input saturation amplitudes of the system are $\tau_{M,x} = 3.76815 \times 10^2 (\text{KN})$, $\tau_{M,y} = 6.80725 \times 10^2 (\text{KN})$ and $\tau_{M,\psi} = 7.3119 \times 10^3 (\text{KNm})$.

The selection of the design parameters of the error performance index functions $\rho_{M,k}(t), \rho_{m,k}(t)$ are shown in Table 1.

**Table 1.** The design parameters of $\rho_{M,k}(t), \rho_{m,k}(t)$.

| | $\text{\ae}_k$ | $\text{\ae}_{M,k}(0)$ | $\text{\ae}_{M,k}(\infty)$ | $\text{\ae}_{m,k}(0)$ | $\text{\ae}_{m,k}(\infty)$ | $\bar{}_k$ |
|---|---|---|---|---|---|---|
| $x$ | 1 | 25.5 | 0.5 | 1.5 | 0.5 | $10^{-2}$ |
| $y$ | 1 | 25.5 | 0.5 | 1.5 | 0.5 | $10^{-2}$ |
| $\psi$ | $\frac{\pi}{180}$ | 15.5 | 0.5 | 1.5 | 0.5 | $10^{-2}$ |

Set the desired value of the ship's DP as $\eta_d = [0, 0, 0]^T$, and select the initial simulation conditions as $\eta(0) = [20\text{m}, 20\text{m}, \pi/18]^T, v(0) = [0, 0, 0]^T$.

The DP control law design parameters are chosen as $a = 0.5; b = 2; T = 3.9$; $C_1 = diag(10^{-3}, 10^{-3}, 10^{-3})$; $K_1 = 10^{-3} \times diag(4, 3, 0.1)$; $K_2 = diag(100, 100, 100)$; $K_0 = diag(20, 20, 20)$; $\Psi = diag(100, 100, 1000)$; $C_3 = 10^5 \times diag(7.2, 12, 9000)$; $C_2 = diag(11, 12, 8); \kappa = diag(10^3, 10^3, 10^3)$.

The state feedback control law $\tau_f$ addresses the control problem of the ship's DP with external disturbances, but does not consider the error constraints and input saturation, and which is designed as

$$\begin{cases} \tau_f = Dv + M\dot{v}_c - L_2 z_2 - \Xi\hat{d}^* \\ \dot{\hat{d}}^* = \Gamma\left[\Xi z_2 - \Lambda\left(\hat{d}^* - d^0\right)\right] \\ T_c\dot{v}_c + v_c = \alpha_c, v_c(0) = \alpha_c(0) \\ \alpha_c = -J^T(\psi)K_c e \end{cases}, \tag{54}$$

where $\Xi = diag(\tanh(z_{2,1}/\varepsilon_1), \tanh(z_{2,2}/\varepsilon_2), \tanh(z_{2,3}/\varepsilon_3)); d^0 = [100, 100, 100]^T$; $L_2 = diag(10^6, 10^6, 10^9); \Gamma = diag(10^3, 10^3, 3 \times 10^5); \Lambda = diag(10^{-7}, 10^{-7}, 10^{-12}); T_c = 10$; $\varepsilon_1 = \varepsilon_2 = 0.05, \varepsilon_3 = 0.001; K_c = 10^{-2} \times diag(8, 8, 8)$.

The robust nonlinear DP control law $\tau_c$ is designed as

$$\tau_c = -K_2 S_2 - Dv + M\frac{\phi_1 - X_d}{T_a} + K_s\xi - \hat{d}, \tag{55}$$

the detailed design process of $\tau_c$ and its design parameters and matrices can be found in [23], and it addresses the control problem of the ship with external disturbances and input saturation.

### 5.1. Case 1. Without Disturbances

The simulation results of the supply ship without disturbances are shown in Figures 4–9. Figures 4 and 5 show that the control laws $\tau$, $\tau_c$ and $\tau_f$ can enable the ship to accurately complete the DP task, and all the ship's positioning errors are within the error constraint range. Figure 6 indicates that the ship's velocities are bounded. From Figure 7, we can see that the output forces and moment of $\tau$ does not exceed the input saturation amplitudes of the propulsion system, which is in line with the practical engineering. This is because the variable gain function can dynamically adjust the control gain according to the ship's positioning errors to limit the output forces and moment of the control law within the input saturation amplitude range. The output forces and moment of the control law $\tau_f$ exceed the saturation amplitude, which is unacceptable in practical engineering. Figure 8 reveals the control performance $e_\eta = \|\eta - \eta_d\|$ of the DP system. It can be clearly seen that all the ship's positioning errors can converge to 0 within a finite time. In summary, DP systems under different control laws have good control performance when the supply ship is not affected by external disturbances.

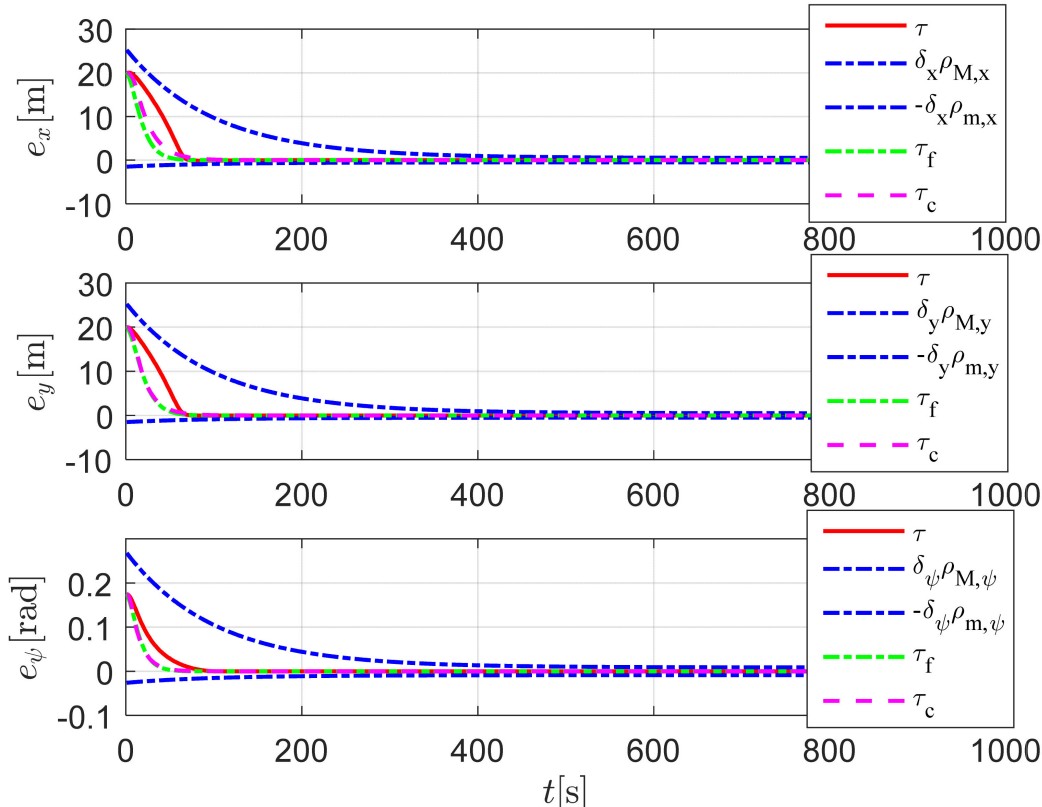

**Figure 4.** The time response curves of positioning error in Case 1.

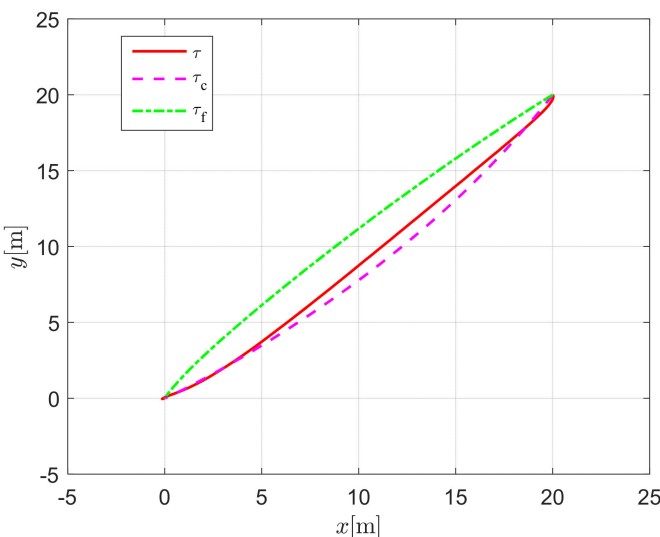

**Figure 5.** The change curves of ship's positioning in Case 1.

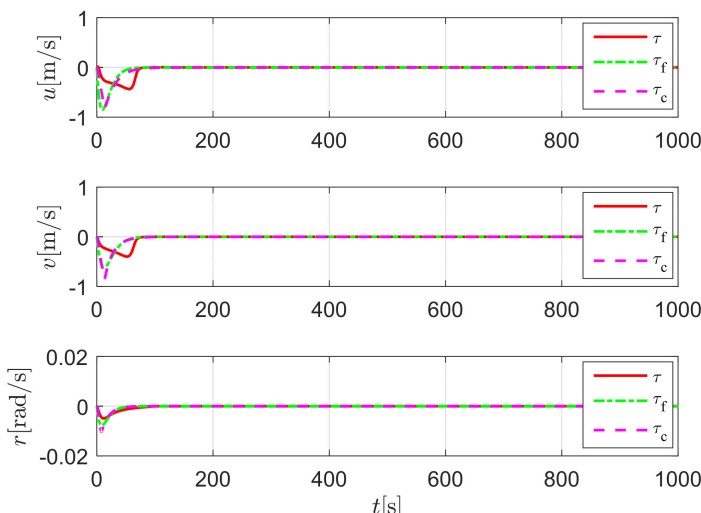

**Figure 6.** The time response curves of ship's velocities in Case 1.

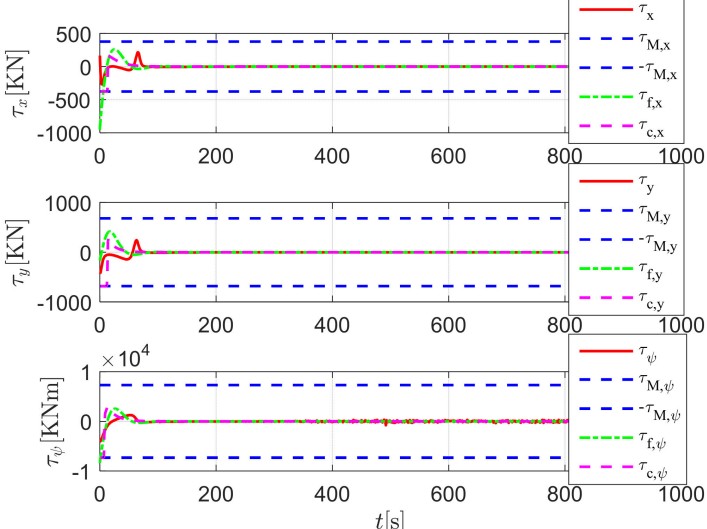

**Figure 7.** The time response curves of the output of the control laws in Case 1.

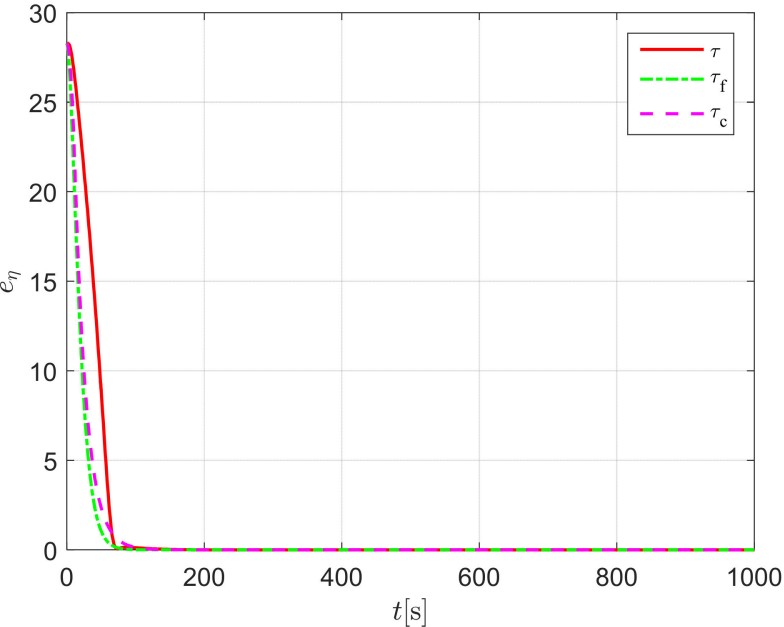

**Figure 8.** The time response curves of the control performance in Case 1.

*5.2. Case 2. With Disturbances*

In order to verify the resistance of the DP system to external disturbances and the non-fragility of the proposed control law to the perturbation of the system's parameters, we will simulate and analyze the DP task of the supply ship with external disturbances in this case. The desired value, initial conditions, design parameters and matrices are set consistent with Case 1.

The external disturbance vector is selected as

$$d = J^T(\psi)b, \tag{56}$$

where $b \in \Re^{3\times1}$ is the disturbance vector in the body-fixed frame and it comes from the first order Markov process

$$\dot{b} = -\kappa^{-1}b + \Psi\theta, \tag{57}$$

where $\kappa = diag(10^3, 10^3, 10^3)$ is the time constant matrix; $\theta \in \Re^{3\times1}$ is a zero-mean Gaussian white noise vector; $\Psi = diag(100, 100, 1000)$ is the amplitude matrix of $\theta$. The initial simulation conditions are selected as $b(0) = [10KN, 10KN, 10KNm]^T$.

The simulation results are shown in Figures 9–15. It can be seen from Figures 9 and 10 that control laws $\tau$ and $\tau_c$ can still enable the supply ship to accurately complete the DP task and make the positioning errors meet the prescribed performance requirements. Figure 11 indicates that the ship's velocities are still bounded. Figures 12 and 13 are estimation curves of different DP control laws for external disturbances. The nonlinear disturbance observer in $\tau$ and $\tau_c$ can accurately estimate the external disturbances, thereby reducing the energy consumption of the ship during the DP process. This also implies that the introduced nonlinear disturbance observer in this paper has a strong adaptive ability, while $\tau_f$ cannot deal with the external disturbances accurately. Figure 14 shows that the control output amplitudes of control laws $\tau$ and $\tau_c$ can still remain within the saturation constraint range. Since the control law $\tau_f$ cannot accurately approximate the external disturbances, the output amplitude of the control law is larger than that of the Case 1, which is still unacceptable. Figure 15 reveals that the proposed DP control law can make the system have better dynamic quality and steady-state performance.

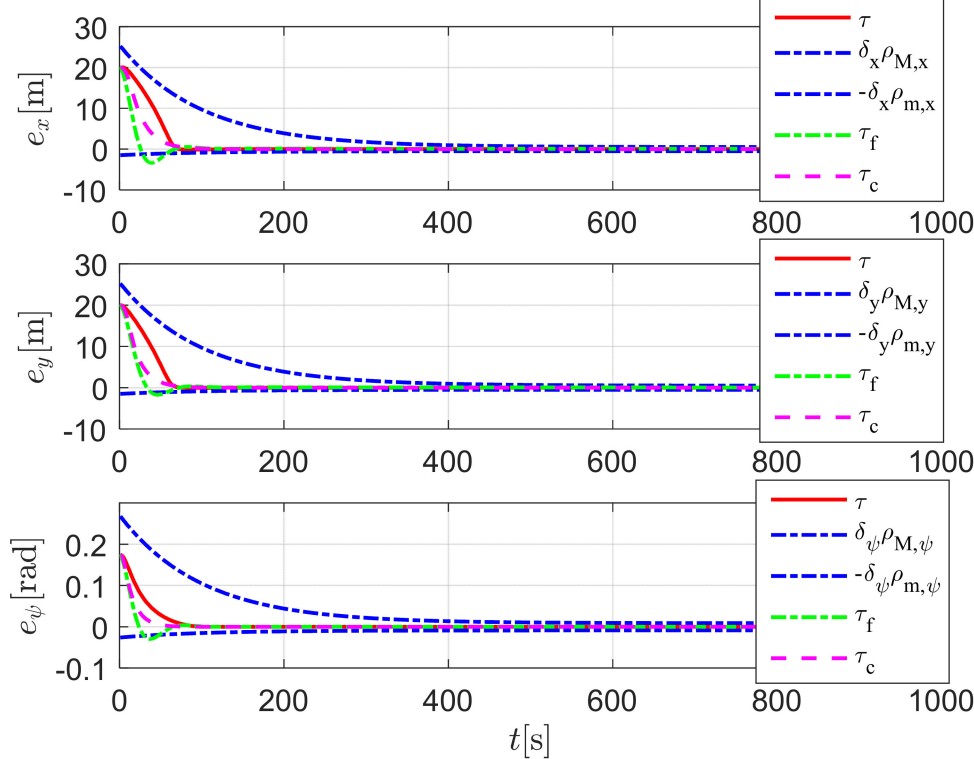

**Figure 9.** The time response curves of positioning error in Case 2.

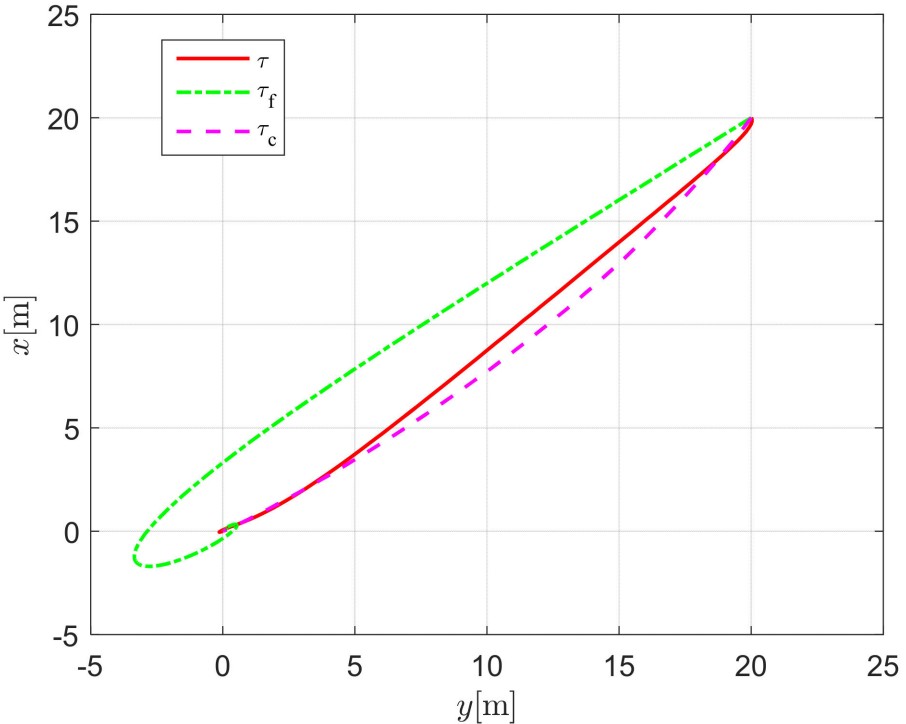

**Figure 10.** The time response curves of positioning error in Case 2.

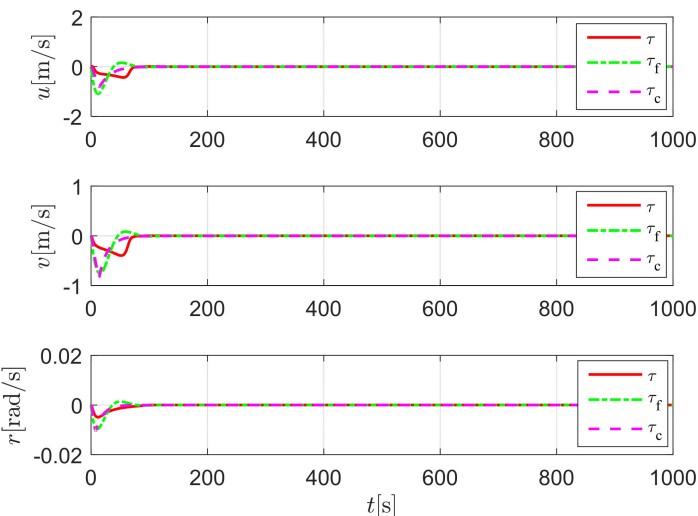

**Figure 11.** The time response curves of ship's velocities in Case 2.

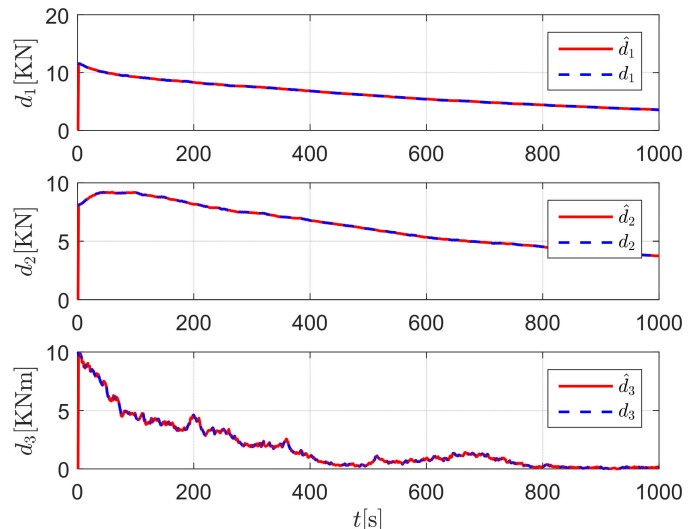

**Figure 12.** The estimation curves of $\tau$ and $\tau_c$ for external disturbances.

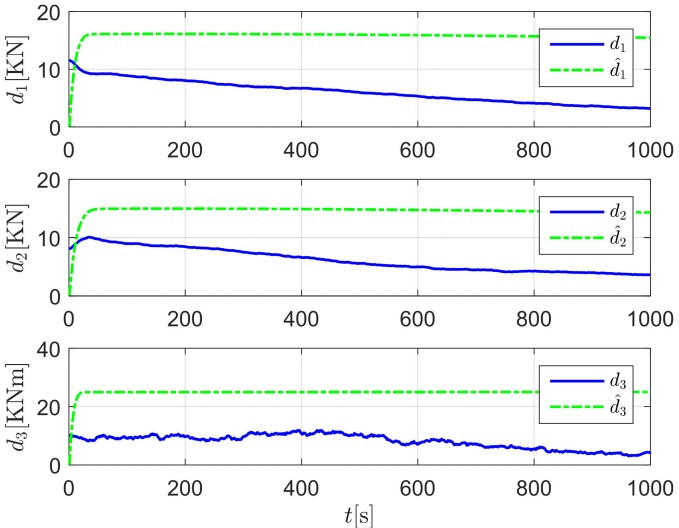

**Figure 13.** The estimation curves of $\tau_f$ for external disturbances.

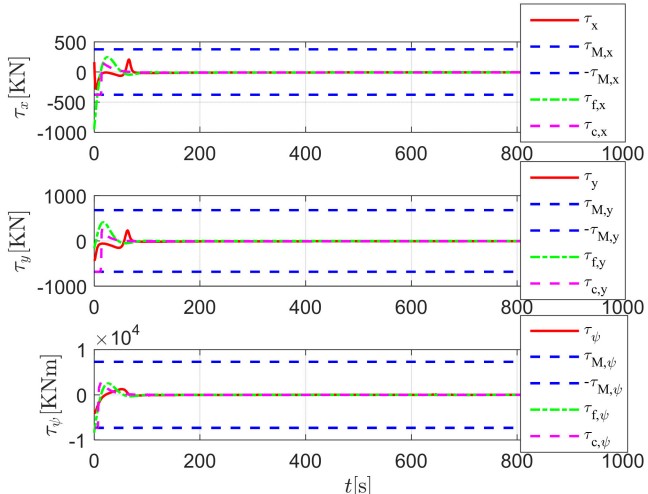

**Figure 14.** The time response curves of the output of the control laws in Case 1.

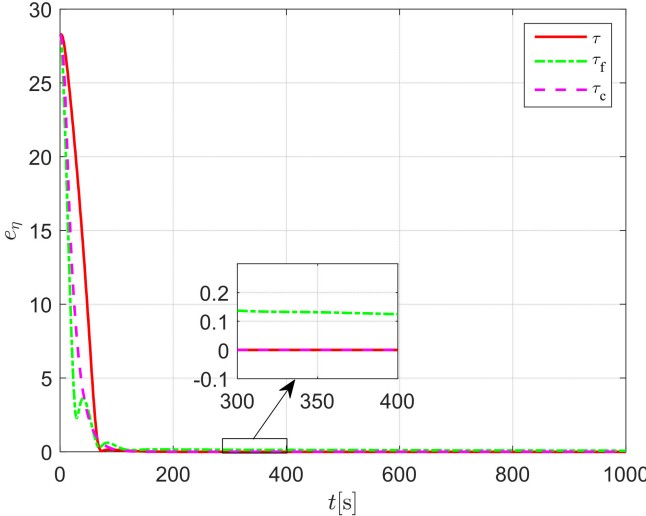

**Figure 15.** The time response curves of the control performance in Case 2.

The analysis of the simulation results of Case 1 and Case 2 illustrates that the DP system has strong resistance to external disturbances, and the proposed DP control law has strong non-fragility to the perturbation of system parameters.

## 6. Conclusions

In the presence of external disturbances positioning error constraints and input saturation, a variable gain prescribed performance DP control law is proposed for DP ships in this paper. A nonlinear disturbance observer is introduced to estimate and compensate the unknown external disturbances. The error performance index functions and error mapping functions are designed to make the ship's DP meet the prescribed performance requirements. A variable gain recursive sliding-mode DSC control is designed to avoid input saturation of the propulsion system and enhance the non-fragility of the control law. The stability of the DP closed-loop control system and the uniformly ultimately boundedness of all signals in the DP control system are proved by the Lyapunov method. Finally, simulation analyses of the DP task of a supply ship are carried out. It illustrated that the proposed control law has strong non-fragility to the perturbation of system's parameters, which can make the DP control system have a strong resistance to external disturbances and ensure that the positioning errors meet the prescribed performance requirements.

The main innovations of this paper are as follows:

(1) An improved DP control law is proposed to resolve the ship's DP problem of external disturbances, input saturation and error constraints, and ensure the uniformity boundness of all signals in the DP closed-loop system.

(2) A prescribed performance control strategy is designed to guarantee that the ship's positioning error meets the preset performance requirements, and a variable gain function with the characteristics of "large gains for small errors, small gains for large errors" is designed to effectively adjust the control gain of the DP control law according to the ship positioning error to ensure that the system does not have input saturation.

(3) Different from the existing DP control laws in [23] and [37], the proposed DP control law in this paper does not need to design the auxiliary dynamic system to handle input saturation and it can make the ships complete the high-performance DP with prescribed performance. Moreover, the proposed control law in this paper not only conforms to the practical ship's DP, but also can be easily extended to the control of other systems with Euler-Lagrange dynamic equations, such as aircraft systems, robotic systems, etc.

The disadvantage of the variable gain recursive sliding-mode DP control law is that the number of control design parameters and matrices increase, while the selection of design parameters and matrices are based on trial and error, which may not be optimal. In the future research, the design matrices and parameters of the variable gain recursive sliding-mode DP control law will be identified and optimized to achieve the optimal control performance, and the unknown kinematics and dynamics mathematical model parameters and unmeasurable velocities of the ship should also be handled in the prescribed performance DP control framework.

**Author Contributions:** Conceptualization, C.G. and Y.S.; methodology, C.G.; software, C.G.; validation, C.G., Y.S. and D.Z.; formal analysis, Y.S.; investigation, D.Z.; resources, C.G.; data curation, C.G.; writing—original draft preparation, C.G.; writing—review and editing, Y.S.; visualization, Y.S.; supervision, D.Z.; project administration, Y.S.; funding acquisition, Y.S. All authors have read and agreed to the published version of the manuscript.

**Funding:** This research received no external funding.

**Acknowledgments:** The authors would like to thank the anonymous reviewers for their useful comments and suggestions.

**Conflicts of Interest:** The authors declare no conflict of interest.

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
