# Peer review of "Variable Gain Prescribed Performance Control for Dynamic Positioning of Ships with Positioning Error Constraints"

_jmse, doi:10.3390/jmse10010074_

Round 1

Reviewer 1 Report

The problem proposed in this paper is generally interesting, and the obtained results are promising and correct. Nevertheless, there are some problems that should be addressed:

1/ The motivation and background of wide practical use of the theoretic results presented should be clearly emphasized to facilitate the readers.

2/ If the Lyapunov functions are chosen via the viewpoint of practical application, the authors should give some effective suggestions. More discussions should be given to clearly demonstrate the limitations/validity of the obtained results.

3/ In the simulation results, discussions and explanations should be provided more on the comparison between both methods. More discussions should be given to clearly demonstrate the effectiveness of the obtained results. Also, the authors should describe the different conditions between case 1 and case 2. The title of sections 5.1 and 5.2 may be changed as Case 1: Without disturbance and Case 2: With disturbance.

4/ The authors only simulate the controller, and no implementation in the actual vehicle is done. Consequently, it is tough to assess if such excellent properties of the controller are needed. From the simulations with satisfactory results, the system performance is expected in actual experiments with your proposed method. The analysis in this paper should be supported by experimental results. The authors should use practical systems to validate the proposed methods with experiment results. This paper now is difficult to prove the advantages of the proposed algorithm.

5/ The controller is compared only with one another. More comparisons to other solutions are welcomed.

6/ Besides, some new relative references have to be cited in relation with the current work on DP control and sliding mode control of marine systems: Station-Keeping Control of a Hovering Over-Actuated Autonomous Underwater Vehicle Under Ocean Current Effects and Model Uncertainties in Horizontal Plane (IEEE Access, 2021); Design of a Non-Singular Adaptive Integral-Type Finite Time Tracking Control for Nonlinear Systems With External Disturbances (IEEE Access, 2021); Perturbation Observer-Based Robust Control Using a Multiple Sliding Surfaces for Nonlinear Systems with Influences of Matched and Unmatched Uncertainties (Mathematics, 2020), Robust Position Control of an Over-actuated Underwater Vehicle under Model Uncertainties and Ocean Current Effects Using Dynamic Sliding Mode Surface and Optimal Allocation Control (Sensors, 2021).

7/ Rewrite the conclusion part since the first paragraph is quite similar to the abstract section.

8/ English needs to be polished. The manuscript should be formatted better and some spelling and grammar should be checked carefully. There are some unclear sentences along  the paper.

Reviewer 2 Report

The article “Variable Gain Prescribed Performance Control for Dynamic Positioning of Ships with Positioning Error Constraints” is based on the simulative work for ships. This work is well written and can be considered for potential publications after the following changes:

1) Briefly explain the novelty of this work

2) Had these simulations with such assumptions are possible to be practically utilized, if yes kindly add the references

 3) What are the gaps left and the authors will like to propose for new researchers to overcome for the continuation of this work

Round 2

Reviewer 1 Report

Thank you for the revised manuscript. I appreciate your efforts to revise the manuscript in light of the comments addressed in the previous review. The new version is good now. The quality of the paper has been improved by properly addressed my previous comments. For this, the paper is much better structured and easy to understand.

Other minor comments:

The format of references needs to be unified with the aim to satisfy the requirement of the journal, and the DOI number of some new references needs to be added in this paper. Please check carefully.

For example, In ref [3], the author´s name is wrong. Please re-arrange as below:

Alattas, K.A., Mobayen, A., Din, S.U., Asad, J.H., Fekih, A., Assawinchaichote, W., Vu, M.T., 2020. Design of a Non-Singular Adaptive Integral-Type Finite Time Tracking Control for Nonlinear Systems With External Disturbances. IEEE Access, Volume 9, pp. 102091-102103. DOI: 10.1109/ACCESS.2021.3098327
